

# Bridging Data Assimilation and Control: Ensemble Model Predictive Control for High-Dimensional Nonlinear Systems

Kenta Kurosawa[a] , Atsushi Okazaki[a,b] , Fumitoshi Kawasaki[c] , Shunji Kotsuki[a,b,d]

[a] *Center for Environmental Remote Sensing, Chiba University, Chiba, Japan*

[b] *Institute for Advanced Academic Research, Chiba University, Chiba, Japan*

[c] *Graduate School of Science and Engineering, Chiba University, Chiba, Japan*

[d] *Research Institute of Disaster Medicine, Chiba University, Chiba, Japan*

*Corresponding author*: Kenta Kurosawa, kurosawa@chiba-u.jp





ABSTRACT:   Model predictive control (MPC) is an optimization-based control framework for linear and nonlinear systems. MPC estimates control inputs by iterative optimization of a cost function that minimizes deviations from a desired state while accounting for control costs over a finite prediction horizon. This process typically involves direct computations in state space through full model evaluations, making it computationally expensive for high-dimensional nonlinear systems. This study introduces ensemble model predictive control (EnMPC), a novel framework for nonlinear control that combines MPC and ensemble data assimilation. EnMPC directly solves the MPC cost function using ensemble smoother methods, including the four-dimensional ensemble variational assimilation method, ensemble Kalman smoother, and particle smoother. By assimilating pseudo-observations that incorporate information about reference trajectories and constraints, EnMPC mitigates nonlinearity and uncertainty in high-dimensional systems, outperforming conventional MPC in computational efficiency through ensemble approximations. In addition, EnMPC is able to determine optimal weights for control inputs by using the analysis error covariance derived from ensemble data assimilation. We present two different approaches for defining control objectives. The penalty term approach applies penalties when model predictions violate pre-defined constraints by assimilating constraint information as pseudo-observations. In contrast, the trajectory tracking approach assimilates pseudo-observations derived from a reference trajectory to lead the system in the direction of the desired state. We perform numerical experiments with idealized models that capture the chaotic nature of atmospheric systems to show that EnMPC efficiently controls the system and offers flexibility for a variety of control objectives.



## 1. Introduction

The intensification of extreme weather events induced by global warming is causing significant damage to human life and property worldwide. As the IPCC sixth assessment report points out, rising temperatures increase the threat by increasing the frequency of heat waves, heavy rains and floods, and the intensity of hurricanes and typhoons (IPCC 2021). The demand for new technological advances is growing as it becomes more difficult to manage the increasing number of extreme weather events with only infrastructure improvements. Since the mid-20th century, researchers have considered interventions such as cloud seeding, where they use silver iodide to induce rainfall. However, while scientific studies have provided evidence to support the effectiveness of the approach to some extent (Langmuir 1948; Ryan and King 1997; Silverman 2001), its efficiency and optimization remain areas of active research.

Model predictive control (MPC) is a powerful control technique that uses dynamic models to predict future behavior and optimize control actions over a finite time horizon (Morari and Lee 1999; Rockett and Hathway 2017; Babu et al. 2019; Schwenzer et al. 2021). As computational power has advanced, the range of its applications has expanded, and new challenges, such as weather control, have become increasingly realistic. However, meteorological systems are highly complex, consisting of numerous interconnected elements such as the atmosphere, oceans, land, and biosphere (Lea et al. 2015; Sluka et al. 2016; Kurosawa et al. 2023). As its behavior exhibits significant nonlinearities, small variations can have unpredictable effects on the entire system (Slingo and Palmer 2011), and the system responds slowly to interventions (Leith 1974), making accurate predictions and control difficult. Moreover, weather models often require significant computational resources due to their high dimensionality and the need for fine temporal and spatial resolutions. Given these characteristics of weather systems, proper handling of uncertainty and the heavy computational cost of calculating optimal control inputs are key challenges for achieving effective weather control.

To properly handle uncertainty, data assimilation integrates observations and numerical models to more accurately estimate the state of the system, and it is widely used in weather forecasting (Houtekamer and Mitchell 1998; Kalnay 2003; Leutbecher and Palmer 2008; Evensen 2009). Miyoshi and Sun (2022) proposed a new experimental framework to systematically evaluate control approaches through ensemble prediction. In the framework, known as the control simulation exper-





iment (CSE), they used ensemble data assimilation for state estimation. Subsequently, Kawasaki and Kotsuki (2024) integrated a conventional MPC method and achieved efficient control with minimal input within the CSE framework. However, the computational cost of calculating optimal control inputs remains high, and there is a need to develop more efficient control methods.

Sawada (2024a,b) proposed a weather control method that combines ensemble data assimilation and MPC, utilizing the ensemble Kalman filter (EnKF) and ensemble Kalman smoother (EnKS) to solve the MPC problem efficiently. Traditional MPC requires direct computations in state spaces and explicit calculation of system evolution within the prediction horizon, whereas ensemble approximations use statistical representations, enabling more efficient control of complex systems. The EnKF-based control method, which directly utilizes the existing EnKF architecture, offers flexibility for geoscience applications but still faces several challenges. First, when calculating the optimal control inputs, the system's behavior within the evaluation horizon or window of the cost function is assumed to be approximately linear. In systems with strong nonlinearity, this approximation does not hold, and errors are likely to occur when calculating the optimal control input (Zhang et al. 2009; Kurosawa and Poterjoy 2021). Second, as used in Sawada (2024a), many control problems commonly add penalty terms to the cost function to handle constraint violations in control objectives. In the penalty-based approaches, when control objectives are unclear or multiple objectives must be balanced, designing the cost function and setting penalties becomes challenging, potentially reducing performance and causing unintended behavior.

To address these challenges, the current study extends the methodology of using ensemble data assimilation for solving MPC problems, building upon the insights of Sawada (2024a). Specifically, we propose an ensemble model predictive control (EnMPC) framework that employs various ensemble data assimilation techniques, including 4D-ensemble-Var (4DEnVar), particle filter (PF), and particle smoother (PS). This approach expands the range of tools available for solving MPC problems in high-dimensional nonlinear systems. As part of this framework, the EnMPC includes the method proposed by Sawada (2024a), which uses the EnKF and EnKS to solve MPC problems. Furthermore, the EnMPC framework introduces not only the penalty-based approach but also a trajectory-tracking approach to achieve control, providing greater flexibility in addressing diverse control objectives. To demonstrate the effectiveness of the proposed EnMPC framework, we conduct a comparison with conventional MPC approaches.





The manuscript is organized in the following manner. Section 2 provides a brief overview of ensemble data assimilation and MPC. We introduce EnMPC in Sec. 3, and Sec. 4 describes the experimental setup. Section 5 presents the experimental results, and the last section concludes the paper with a summary of the key findings, potential applications, and directions for future research.

## 2. MPC and data assimilation

This section provides a brief overview of MPC and ensemble data assimilation, which constitute the proposed EnMPC framework. We begin by presenting the MPC algorithm for dealing with control problems. Subsequently, we outline ensemble data assimilation, focusing on 4DEnVar, EnKF, and PF. This section explains MPC and data assimilation individually, while Sec. 3 highlights their similarities, differences, and how they are combined to form EnMPC.

*a. MPC*

MPC is a control strategy that optimizes control inputs by using a dynamic model to predict the future behavior of the system. MPC solves an optimization problem at each time step to minimize a cost function over a finite predictive horizon. The specific design of the cost function depends on the application, but the general formulation can be expressed as:

$$J(\mathbf{u}_0, \mathbf{u}_1, \ldots, \mathbf{u}_{T_c}) = \underbrace{\sum_{t=0}^{T_c} \mathbf{u}_t^\top \mathbf{C}^{u^{-1}} \mathbf{u}_t}_{J_{\text{input}}} + \underbrace{\sum_{t=0}^{T_p} (\mathbf{r}_t - H^c(\mathbf{x}_t))^\top \mathbf{C}^{r^{-1}} (\mathbf{r}_t - H^c(\mathbf{x}_t))}_{J_{\text{state}}}.$$

$$\text{s.t.} \quad \mathbf{x}_{t+1} = M_t(\mathbf{x}_t, \mathbf{u}_t). \tag{1}$$

Here, $\mathbf{x}_t$ denotes the state variable at time $t$. The next state $\mathbf{x}_{t+1}$ is obtained by integrating the nonlinear forecast model operator $M_t$ forward from the current state $\mathbf{x}_t$ and the control input $\mathbf{u}_t$. The control input cost $J_{\text{input}}$ is typically optimized over a shorter control horizon $T_c$ within the prediction horizon $T_p$. $J_{\text{input}}$ penalizes the magnitude of the control input, preventing it from being excessively large. The state deviation cost $J_{\text{state}}$ evaluates the difference between model-predicted states and the control objective $\mathbf{r}$, and the optimization problem is performed over a finite prediction horizon $T_p$. $H^c$ is an operator that maps the state variables $\mathbf{x}$ to the control variables. $\mathbf{C}^u$ and





$\mathbf{C}^r$ are weighting matrices for the control input $\mathbf{u}$ and the deviations between state variables and control objective, respectively.

Among the two components of the cost function in (1), the state deviation cost $J_{state}$ typically has the highest computational cost. This is because it involves predicting and evaluating the future states of the system over the entire prediction horizon, which requires extensive computations, especially for complex or nonlinear systems. The ensemble approximation can mitigate this computational cost by using representative trajectories to approximate future states, as discussed in Sec. 3.

*b. The four-dimensional variational method (4DVar) and 4DEnVar*

The 4DVar method estimates the optimal initial state $\mathbf{x}_0$ over a time window by considering the misfits between observations and forecast model states at multiple times. This process is achieved by minimizing the following cost function (Talagrand 2014; Bannister 2017):

$$J(\mathbf{x}_0) = \underbrace{\left(\mathbf{x}_0 - \mathbf{x}_0^b\right)^\top \mathbf{B}^{-1} \left(\mathbf{x}_0 - \mathbf{x}_0^b\right)}_{J_{\text{background}}} + \underbrace{\sum_{t=0}^{\tau} \left(\mathbf{y}_t - H(\mathbf{x}_t)\right)^\top \mathbf{R}^{-1} \left(\mathbf{y}_t - H(\mathbf{x}_t)\right)}_{J_{\text{observation}}},$$

$$\text{s.t.} \quad \mathbf{x}_{t+1} = M_t(\mathbf{x}_t)$$

(2)

The first term in (2) qualifies the difference between the initial guess (background or prior) $\mathbf{x}_0^b$ and the estimated state $\mathbf{x}_0$, weighted by the background error covariance matrix $\mathbf{B}$. The second term in (2) measures the misfit between the state variables and the observations $\mathbf{y}$ at times $t = 0, 1, 2, ..., \tau$. The observation operator $H$ maps the state $\mathbf{x}$ to the observation space, and $\mathbf{R}$ represents the observation error covariance matrix. The time window $\tau$ is referred to as the data assimilation window and plays the same role as the prediction horizon $T_p$ in MPC. Therefore, the second term $J_{\text{observation}}$ in (2) serves a similar purpose to the state deviation cost $J_{\text{state}}$ in the MPC cost function (1), as both evaluate the discrepancies between the predicted states and the target values or observations over a specific time horizon.

Operational systems often implement 4DVar using an incremental approach to utilize the linearized model instead of the full nonlinear model (Courtier et al. 1994). Defining $\delta\mathbf{x}_0 = \mathbf{x}_0 - \mathbf{x}_0^b$,





the cost function $J(\mathbf{x}_0)$ in (2) as becomes:

$$J(\delta\mathbf{x}_0) = \underbrace{\delta\mathbf{x}_0^\top \mathbf{B}^{-1}\delta\mathbf{x}_0}_{J_{\text{background}}} + \underbrace{\sum_{t=0}^{\tau}(\mathbf{H}\delta\mathbf{x}_t - \mathbf{d}_t)^\top \mathbf{R}^{-1}(\mathbf{H}\delta\mathbf{x}_t - \mathbf{d}_t)}_{J_{\text{observation}}},$$

$$\text{s.t.} \quad \delta\mathbf{x}_{t+1} = \mathbf{M}_t(\delta\mathbf{x}_t) \tag{3}$$

where $\mathbf{M}_t$ and $\mathbf{H}$ are the tangent linear operators of $M_t$ and $H$, respectively. The innovation vector $\mathbf{d}_t$ is defined as $\mathbf{d}_t = \mathbf{y}_t - H[M_t(\mathbf{x}_0^b)]$.

The convergence rate of the optimization problem depends on the condition number of the Hessian matrix (Zupanski 1996). In operational data assimilation systems using atmospheric models, the dimension of the state vector is typically on the order of $O(10^{10})$ or greater. This results in a background error covariance matrix $\mathbf{B}$ that is too large to be explicitly represented or handle directly. To address this computational challenge, operational systems commonly employ the following approach (Buehner 2005; Wang et al. 2010; Zhu et al. 2022):

$$\begin{cases} \delta\mathbf{x}_0 = \mathbf{U}^x\mathbf{v}, \\ \mathbf{H}\delta\mathbf{x}_t = \mathbf{U}_t^y\mathbf{v}, \end{cases} \tag{4}$$

Here, $\mathbf{U}^x$ is a square root of the background error covariance matrix ($\mathbf{B} = \mathbf{U}^x\mathbf{U}^{x\top}$; Lorenc 2003), and $\mathbf{v}$ is the new control variable in the reduced-dimension space. The initial perturbation $\delta\mathbf{x}_0$ and the observation perturbation $\mathbf{H}\delta\mathbf{x}_t$ are projected onto a subspace spanned by ensemble members using the transformation matrices $\mathbf{U}^x$ and $\mathbf{U}_t^y$, respectively. The perturbation matrices $\mathbf{U}^x$ and $\mathbf{U}^y$ are defined as follows:

$$\begin{cases} \mathbf{U}^x = \frac{1}{\sqrt{N_e-1}}\left[\delta\mathbf{x}^{(1)}, \quad \delta\mathbf{x}^{(2)}, \quad \cdots, \quad \delta\mathbf{x}^{(N_e)}\right] \\[2ex] \mathbf{U}^y = \frac{1}{\sqrt{N_e-1}}\left[\delta\mathbf{y}^{(1)}, \quad \delta\mathbf{y}^{(2)}, \quad \cdots, \quad \delta\mathbf{y}^{(N_e)}\right], \end{cases} \tag{5}$$

where $N_e$ is the ensemble size, $\delta\mathbf{x}^{(k)}$ and $\delta\mathbf{y}^{(k)}$ are the $k$-th ensemble perturbations for the model state and observation space, respectively. Perturbations in observation space are calculated using the tangent linear observation operator, where $\delta\mathbf{y} = \mathbf{H}\delta\mathbf{x}$. By adopting this transformation, the cost





function is reformulated as:

$$J(\mathbf{v}) = \underbrace{\mathbf{v}^\top \mathbf{v}}_{J_{\text{background}}} + \underbrace{\sum_{t=0}^{\tau} (\mathbf{U}_t^y \mathbf{v} - \mathbf{d}_t)^\top \mathbf{R}^{-1} (\mathbf{U}_t^y \mathbf{v} - \mathbf{d}_t)}_{J_{\text{observation}}}. \tag{6}$$

To minimize (6), $\mathbf{v}$ must satisfy the condition $(\partial J / \partial \mathbf{v})^\top = \mathbf{0}$. As a result, this approach eliminates the need for an adjoint model, as all calculations occur within the subspace spanned by the ensemble samples. This incremental 4DEnVar approach combines with ensemble-based transformations thus balances computational efficiency and the practical constraints of high-dimensional data assimilation systems. For further details on these methods, we encourage readers to review the mathematical descriptions in Liu et al. (2009), Fairbairn et al. (2014), Poterjoy and Zhang (2015), and Kurosawa and Poterjoy (2021).

### c. EnKF and EnKS

In this study, the control method based on the EnKF adopts the framework proposed in Sawada (2024a). The EnKF minimizes the following cost function to obtain the analysis state:

$$J(\mathbf{x}_0) = \underbrace{(\mathbf{x}_0 - \overline{\mathbf{x}_0^b})^\top \mathbf{P}^{b^{-1}} (\mathbf{x}_0 - \overline{\mathbf{x}_0^b})}_{J_{\text{background}}} + \underbrace{(\mathbf{y}_0 - H(\mathbf{x}_0))^\top \mathbf{R}^{-1} (\mathbf{y}_0 - H(\mathbf{x}_0))}_{J_{\text{observation}}}. \tag{7}$$

Here, $\overline{\mathbf{x}_0^b}$ is the ensemble mean of the background state variables and $\mathbf{P}^b$ represents the background error covariance matrix. As in 4DVar, MPC and EnKF consider similar cost components, taking into account the background information and discrepancies in their respective frameworks.

The EnKF efficiently reduces the computational cost by representing the error covariance matrix $\mathbf{P}^b$ statistically using ensemble members as follows (Evensen 1994; Whitaker and Hamill 2002; Houtekamer and Zhang 2016):

$$\mathbf{P}^b = \mathbf{E}\mathbf{E}^{\mathrm{T}}, \tag{8}$$

$$\mathbf{E} = \frac{1}{\sqrt{N_e - 1}} [\delta \mathbf{x}^{(1)}, \dots, \delta \mathbf{x}^{(N_e)}], \tag{9}$$





where $\mathbf{E}$ is the matrix of ensemble members, with each column representing the perturbation

from the forecast state. Thus, analytically solving the cost function in (7) yields the update of

the ensemble mean. Unlike the variational methods discussed in Sec.2.b, which require iterative

numerical optimization to minimize their respective cost functions, EnKF does not require such

iterations.

Regarding the update of ensemble members, we obtain the ensemble perturbation matrix $\mathbf{X}^a$

using the ensemble transform Kalman filter (ETKF; Bishop et al. 2001; Hunt et al. 2007), as

follows:

$$\mathbf{X}^a = \mathbf{X}^b \mathbf{W}^a, \tag{10}$$

$$\mathbf{W}^a = [(N_e - 1)\tilde{\mathbf{P}}^a]^{1/2}, \tag{11}$$

$$\tilde{\mathbf{P}}^a = [(N_e - 1)\mathbf{I} + (\mathbf{Y}^b)^\top \mathbf{R}^{-1} \mathbf{Y}^b]^{-1}. \tag{12}$$

Here, $\mathbf{X}^b$ is the background perturbations, and $\tilde{\mathbf{P}}^a$ represents the analysis error covariance matrix

in the transformed space. $\mathbf{Y}^b$ represents the perturbation of the background ensemble in the

observation space, and the weights $\mathbf{W}^a$ are then derived based on the analysis covariance. Similarly

to 4DEnVar, which uses ensemble approximations to project initial and observation perturbations

onto a subspace spanned by ensemble members, the ETKF efficiently reduces the dimensionality

of the analysis problem with ensemble-based transformations.

While EnKF is effective for real-time state estimation, EnKS improves estimation accuracy

further by considering observations over a time window and incorporating their influence ret-

rospectively. In this study, we employ a four-dimensional extension of the ETKF, which uses

temporal correlations within the data to achieve more accurate estimation. For a comprehensive

explanation, please refer to Miyoshi and Aranami (2006) and Hunt et al. (2007).

*d. PF and PS*

Variational methods and EnKF estimate the analysis state by assuming Gaussian error statistics

for the background and observations and minimizing the cost functions defined in (2) and (7). In

contrast, the PF does not assume Gaussianity or linearity but approximates the entire probability

distribution of the state as a set of particles (ensembles or samples). By assigning a likelihood to





each particle, PF estimates the analysis state, making it suitable for systems with strong nonlinearity and non-Gaussianity. The particle distribution plays a similar role to the error covariance matrices ($\mathbf{B}$ and $\mathbf{P}$) used in the variational methods and EnKF. Unlike these methods, however, PF does not explicitly calculate the error covariance; instead, the particle distribution implicitly represents their statistical properties of the background error covariance. For each particle $\mathbf{x}^{(k)}$, the likelihood is calculated as:

$$p(\mathbf{y}|\mathbf{x}^{(k)}) \propto \exp\left(-\frac{1}{2}(\mathbf{y}-H(\mathbf{x}^{(k)}))^\top \mathbf{R}^{-1}(\mathbf{y}-H(\mathbf{x}^{(k)}))\right). \tag{13}$$

This calculation resembles the state deviation term $J_{\text{state}}$ in (1) for MPC, where posterior states are penalized based on their deviation from the reference. The likelihoods are normalized to produce the particle weights $\boldsymbol{\lambda}^{(k)}$:

$$\boldsymbol{\lambda}^{(k)} = \frac{p(\mathbf{y}|\mathbf{x}^{(k)})}{\sum_{m=1}^{N_e} p(\mathbf{y}|\mathbf{x}^{(m)})}. \tag{14}$$

Using the weighted particles, PF approximates the posterior distribution (filter distribution) as:

$$p(\mathbf{x}|\mathbf{y}) \approx \sum_{m=1}^{N_e} \boldsymbol{\lambda}^{(m)} \delta(\mathbf{x}-\mathbf{x}^{(m)}), \tag{15}$$

where $\delta(\mathbf{x}-\mathbf{x}^{(k)})$ represents a Dirac delta function centered at particle $\mathbf{x}^{(k)}$. This representation indicates that the posterior distribution is expressed as a discrete set of weighted particles. To better approximate the posterior distribution and mitigate degeneracy, where some particles have negligible weights, a resampling step is performed. During resampling, particles with higher weights are replicated, while those with lower weights are discarded, ensuring the ensemble remains focused on the most likely regions of the state space. This resampling process mirrors the adaptive selection of control inputs in MPC, which focuses optimization efforts on trajectories that minimize the cost function.

The PF is a method for sequentially estimating states, while the PS uses future observation data to provide more accurate state estimates. Applying the weights calculated during the filter update within a data assimilation window, PS uses of the future weights to find the smoother solution at any point throughout the window. This approach is justified by the Markov property, where the system's future evolution depends solely on its current state (Chopin and Papaspiliopoulos 2020;





Nyobe et al. 2023). By taking advantage of this feature, the smoother can produce more accurate estimates over the assimilation window by using future data and previously calculated weights.

We note that several studies propose strategies to address degeneracy and maintain particle diversity (e.g., Penny and Miyoshi 2016; Potthast et al. 2019; Kotsuki et al. 2022). These differences include the resampling strategy, techniques to mitigate particle collapse, and localization to manage high-dimensional systems. The current study adopts the PF and PS algorithm based on the recently proposed PF by Poterjoy (2022), as it employs regularization and iterative updates to effectively address degeneracy and maintain particle diversity. For more detailed information on this approach, please refer to Poterjoy (2016, 2022) and Kurosawa and Poterjoy (2023).

## 3. Ensemble Model Predictive Control

Section 2 provides an overview of conventional MPC and ensemble data assimilation, highlighting their shared goal of determining optimal inputs based on the current state and future predictions. This section introduces a new control technique called EnMPC, which integrates these two methods. Since EnMPC uses the principles of data assimilation, it incorporates pseudo-observations that contain information about constraints and reference trajectories typically used in MPC. These pseudo-observations are assimilated in a manner similar to actual observations in data assimilation, allowing the cost function in EnMPC to adopt a structure similar to that in ensemble data assimilation.

Sawada (2024a) focuses on similarities and differences between EnKF and MPC and introduces EnKF-based EnMPC. Extending this concept, this section focuses on the mathematical formulation of EnMPC, using ideas from 4DVar to develop a 4DEnVar-based EnMPC approach. We define the formulation of EnMPC in a straightforward manner by modifying the MPC cost function in (1) to make it closer in structure to that of 4DEnVar in (6).

First, data assimilation focuses on state estimation by updating the initial conditions for model integration, while MPC estimates control inputs applied during the control horizon $T_c$. The proposed EnMPC framework treats the control inputs as acting only at the initial time, similar to how data assimilation updates the initial states. While this assumption simplifies the framework, extending EnMPC to optimize control inputs over the entire control horizon $T_c$ remains an important direction for future research. Second, the conventional MPC uses the reference vector **r** in (1) to





represent the desired state. In EnMPC, we reformulate the reference vector as a pseudo-observation vector $\mathbf{y}^p$. This allows EnMPC to handle reference information in the same way data assimilation incorporates observations. The cost function for EnMPC is therefore expressed as follows:

$$
J(\mathbf{x}_0) = \underbrace{(\mathbf{x}_0 - \overline{\mathbf{x}_0^a})^\top \mathbf{P}^{a^{-1}} (\mathbf{x}_0 - \overline{\mathbf{x}_0^a})}_{J_{\text{input}}} + \underbrace{\sum_{t=0}^{T_p} (\mathbf{y}_t^p - H^p(\mathbf{x}_t))^\top \mathbf{C}^{r^{-1}} (\mathbf{y}_t^p - H^p(\mathbf{x}_t))}_{J_{\text{state}}}.
$$

$$
\text{s.t.} \quad \mathbf{x}_{t+1} = M_t(\mathbf{x}_t).
$$

(16)

Here, $\mathbf{P}^a$ is the analysis error covariance matrix, as the ensemble updated by data assimilation can be used directly. $H^p$ is the operator that maps the state vector to the pseudo-observation space. As described in Section 2b, applying ensemble approximations to the cost function in (16) yields:

$$
J(\mathbf{v}) = \underbrace{\mathbf{v}^\top \mathbf{v}}_{J_{\text{input}}} + \underbrace{\sum_{t=0}^{T_p} (\mathbf{U}_t^y \mathbf{v} - \mathbf{d}_t^p)^\top \mathbf{C}^{r^{-1}} (\mathbf{U}_t^y \mathbf{v} - \mathbf{d}_t^p)}_{J_{\text{state}}},
$$

(17)

where the innovation vector $\mathbf{d}_t^p$ is defined as $\mathbf{d}_t^p = \mathbf{y}_t - H^p[M_t(\mathbf{x}_0^a)]$. The gradient of the cost function in (17) with respect to $\mathbf{v}$ is expressed as:

$$
\left(\frac{\partial J}{\partial \mathbf{v}}\right)^\top = \mathbf{v} + \sum_{t=0}^{T_p} \mathbf{U}_t^{y\top} \mathbf{C}^{r^{-1}} \left[\mathbf{U}_t^y \mathbf{v} - \mathbf{d}_t^p\right]
$$

(18)

This expression shows that solving the EnMPC optimization problem does not require the full nonlinear model or its tangent linear model, as the ensemble approximations are used to calculate the gradient.

A key feature of EnMPC is its ability to assimilate pseudo-observations in a manner similar to actual observations in data assimilation. Therefore, the EnMPC approach, which directly solves the MPC cost function using ensemble estimations, is not limited to 4DEnVar-based framework, but can also be applied to EnKS- or PS-based frameworks. Moreover, EnMPC offers flexibility in setting control objectives through the use of the pseudo-observations and operators. This study introduces two approaches for defining control objectives. The first, referred to as the "penalty term approach," creates a pseudo-observation vector only when the model prediction exceeds a





predefined threshold, as used in Sawada (2024a). The second, called the "trajectory tracking approach," generates pseudo-observation data directly from the reference trajectory, enabling straightforward objective definition. We provide more details in Sec. 4c. Lastly, EnMPC can appropriately handle sampling errors and uncertainties by incorporating techniques from ensemble data assimilation, such as localization and inflation, as detailed in Sawada (2024b).

## 4. Experimental settings

In this section, we describe the experimental setup used to evaluate the effectiveness of the proposed EnMPC through numerical experiments using the Lorenz63 (Lorenz 1963) model. Our experiments follow the CSE procedure (Miyoshi and Sun 2022; Sun et al. 2023; Ouyang et al. 2023; Kawasaki and Kotsuki 2024; Sawada 2024a).

### a. Experimental procedure

Figure 1 illustrates the process of the CSE using the proposed EnMPC. The procedure consists of the following steps:

1. To obtain an accurate estimate of the current state of the system, we first simulate observations from the nature run (NR; or the true state of the system). We then perform a conventional ensemble data assimilation using these simulated observations, which corresponds to the filter update (Fig. 1a). This step includes estimating unobserved state variables that are targets for control. The outcome of this process provides the initial conditions necessary for the subsequent control step.

2. Based on the state estimated in the previous step, we determine the optimal control input using the proposed EnMPC. We consider two approaches for control input determination:

   (a) Penalty term approach

   This approach uses a pseudo-observation operator, which acts as a penalty function commonly used in the conventional MPC. Pseudo-observations are generated when the model prediction violates the predefined constraints, effectively penalizing unsuitable behavior (Fig. 1b1).



(b) Trajectory tracking approach

In the current study, pseudo-observations are directly derived from the reference tra-
jectory, making it straightforward to guide the system toward the desired state (Fig.
1b2).

3. The optimal control input determined in the second step is applied to the NR to perform the
control, and the state is integrated forward to the next time step. Similarly, we apply the same
control input to the ensemble members and predict their states for the next time step. With the
updated system state and ensemble predictions, we restart the CSE cycle from the first step
(Fig. 1c).

Here, we emphasize that for state estimation in the first step (Fig. 1a), we employ conventional
ensemble data assimilation methods, corresponding to the filter update. In contrast, the second
step (Fig. 1b) utilizes the proposed EnMPC, which is based on an ensemble smoother update, to
determine the optimal control inputs. For data assimilation in the first step, we consistently use the
ETKF, regardless of which ensemble smoother update method (4DEnVar, EnKS, or PS) is employed
in EnMPC in the second step. This uniformity ensures that any differences in performance are
solely due to the choice of method in EnMPC in the second step and not influenced by variations
in the state estimation in the first step. Lastly, the current study adopts a moving horizon window
of one step. That is, regardless of the length of the prediction horizon used in EnMPC, data
assimilation and control input estimation are performed at every time step in each cycle.

*b. Model description*

The current study uses the Lorenz63 (Lorenz 1963) model for testing the proposed control
method. Although relatively simple in structure, the model is widely employed as a testbed for
understanding chaotic system behavior. This study aims to demonstrate the effectiveness of EnMPC
for control and parameter estimation in such chaotic systems.





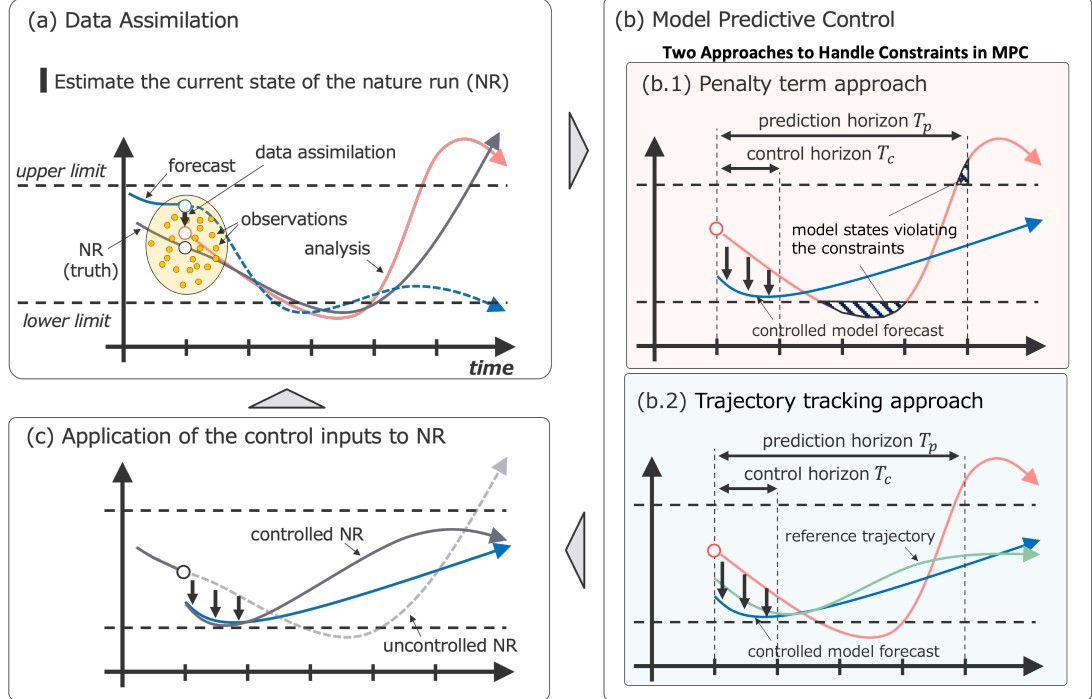

FIG. 1.   Algorithmic flow of the proposed EnMPC-based CSE for a system with upper and lower limits. (a) State estimation: estimates the current state of the system using data assimilation (filter update). (b) Control input optimization: determines the optimal control inputs using the proposed EnMPC framework based on ensemble forecasts; (b.1) penalty term approach and (b.2) trajectory tracking approach. (c) Application of control inputs: applies the optimized control inputs to the NR, integrates the system state forward to the next time step, and returns to the filter update step (a), restarting the CSE cycle.

The Lorenz63 model is a simplified model of atmospheric convection and is represented by the following set of ordinary differential equations with three state variables:

$$
\begin{cases}
\dfrac{dx}{dt} = \sigma(y - x), \\[2mm]
\dfrac{dy}{dt} = x(\rho - z) - y, \\[2mm]
\dfrac{dz}{dt} = xy - \beta z.
\end{cases}
\tag{19}
$$





Following Lorenz (1963), we set system parameters $\sigma = 10$, $\rho = 28$, and $\beta = 8/3$. The time step is set to $\Delta t = 0.01$ (units defined arbitrarily as 1 hour; see Lorenz (1963)). The Lorenz63 model is characterized by its chaotic trajectory, which oscillates around two unstable fixed points, $(\pm\sqrt{72}, \pm\sqrt{72}, 27)^\top$ (Kaiser et al. 2018).

Using the Lorenz63 model, the current study investigates two scenarios for control input estimation: estimating only $u_x$. as shown in (20), and estimating all three control variables $u_x$, $u_y$, and $u_z$, as shown in (21):

$$\begin{cases} \dfrac{dx}{dt} = \sigma(y-x) + u_x, \\ \dfrac{dy}{dt} = x(\rho - z) - y, \\ \dfrac{dz}{dt} = xy - \beta z, \end{cases} \tag{20}$$

and

$$\begin{cases} \dfrac{dx}{dt} = \sigma(y-x) + u_x, \\ \dfrac{dy}{dt} = x(\rho - z) - y + u_y, \\ \dfrac{dz}{dt} = xy - \beta z + u_z. \end{cases} \tag{21}$$

The control objective in the current study is to keep the value of $x$ in the model positive, ensuring that the system avoids undesired negative states. Note that the control inputs are applied to the time derivatives of the state variables, rather than the states themselves.

### c. Pseudo-observations and operators

In the proposed EnMPC framework, we address control problems using two approaches:the penalty term approach and the trajectory tracking approach. Each approach employs different methods for generating pseudo-observations $\mathbf{y}^p$ and operators $H^p$. Throughout our experiments, we set the psuedo-observation error covariance matrix $\mathbf{C}^r$, which acts as the weighting matrix for the deviations between state variables and control objectives, to $\mathbf{C}^r = 0.01\mathbf{I}$, where $\mathbf{I}$ is the identity matrix. This configuration is based on insights from preliminary experiments and the detailed investigation in Sawada (2024a,b).





### 1) PENALTY TERM APPROACH

In the penalty term approach, we generate pseudo-observations to ensure that variables remain within specified thresholds. We set the pseudo-observation value to the threshold and assimilate it into the state space via a pseudo-observation operator. Sawada (2024a) employs a similar strategy, designing the control operator to impose penalties when constraints are violated. This approach effectively makes the pseudo-observation operator serve the same role as the penalty function commonly used in conventional MPC.

The control objective of the current study is to keep the $x$ value positive in the Lorenz63 model. When we apply the penalty term approach for the objective (as detailed in Sec. 5a), we use the following pseudo-observation operator $H^p$:

$$H^p(x) = \frac{\log(1 + \exp(-ax))}{a}, \tag{22}$$

where $a$ is a positive constant that determines the sharpness of the penalty function. As shown in Fig. 2, when $a = 100$, the function approximates a hinge function that activates the penalty only when $x$ becomes less than zero. To keep the value of $x$ non-negative, we set the pseudo-observation $\mathbf{y}^p = 0$. We then use a pseudo-observation operator $H^p$ to project the model state $x$ into the observation space $H^p(x)$, effectively imposing a penalty when $x$ violates the constraint. A smaller $a$ results in a smoother transition, applying penalties even when $x$ is above the threshold but approaching the threshold, as shown in Fig. 2.

Figure 3 illustrates the impact of changing the parameter $a$ in the pseudo-observation operator using the Lorenz63 model. Control input $u_x$ is calculated at each time step using (20), and the prediction horizon $T_p$ is set to 48 steps (= 48 hour). For this demonstration, we use the 4DEnVar-based EnMPC with 10 ensemble members. The parameters for this experiment are summarized in Table 1a.

When $a = 100$, the control inputs are relatively large due to delayed activation of the penalty term, resulting in spike-like control behavior (Fig. 3a,d). Decreasing the value of $a$ activates the penalty more gradually, allowing the control to respond earlier, thus preventing $x < 0$ more smoothly (Figs. 3b–c and e–f). These results show that the choice of $a$ is critical and depends on the specific control objectives. When the control objective is to maintain the system state close to the threshold, a




larger $a$ may be necessary, leading to larger and abrupt control inputs. On the other hand, when

staying further from the threshold is acceptable, a smaller $a$ can reduce the overall control inputs,

although the model states may not closely approach the threshold. This highlights the importance

of selecting an appropriate pseudo-observation operator to balance the desired control objectives

with the acceptable magnitude of control inputs.

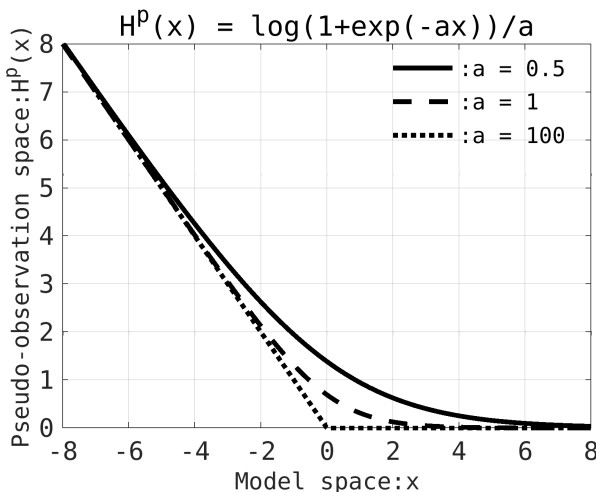

FIG. 2. Comparison of the pseudo-observation operator $H^p(x) = \log(1 + \exp(-ax))/a$ used in this study for

different values of the positive constant parameter $a$. The solid line, dashed line, and dotted line represent the

cases where $a = 0.5$, $a = 1$, and $a = 100$, respectively. The horizontal axis represents values in the model space,

while the vertical axis represents the values projected into the pseudo-observation space using the operator.

2) TRAJECTORY TRACKING APPROACH

In the trajectory tracking approach, the current study first defines a reference trajectory that

satisfies the desired constraints. We then control or guide the system to follow this trajectory by

assimilating pseudo-observations. The pseudo-observations are generated by taking the states of

the reference trajectory at each observation time.

For the experiment using the Lorenz63 model (as detailed in Sec. 5b), we use the trajectory

generated by Kawasaki and Kotsuki (2024) as the reference. This trajectory satisfies the constraint

$x > 0$ and is obtained using conventional MPC by applying control inputs $u_x$, $u_y$, and $u_z$ to the

Lorenz63 model. We generate the pseudo-observations from the reference every time step for



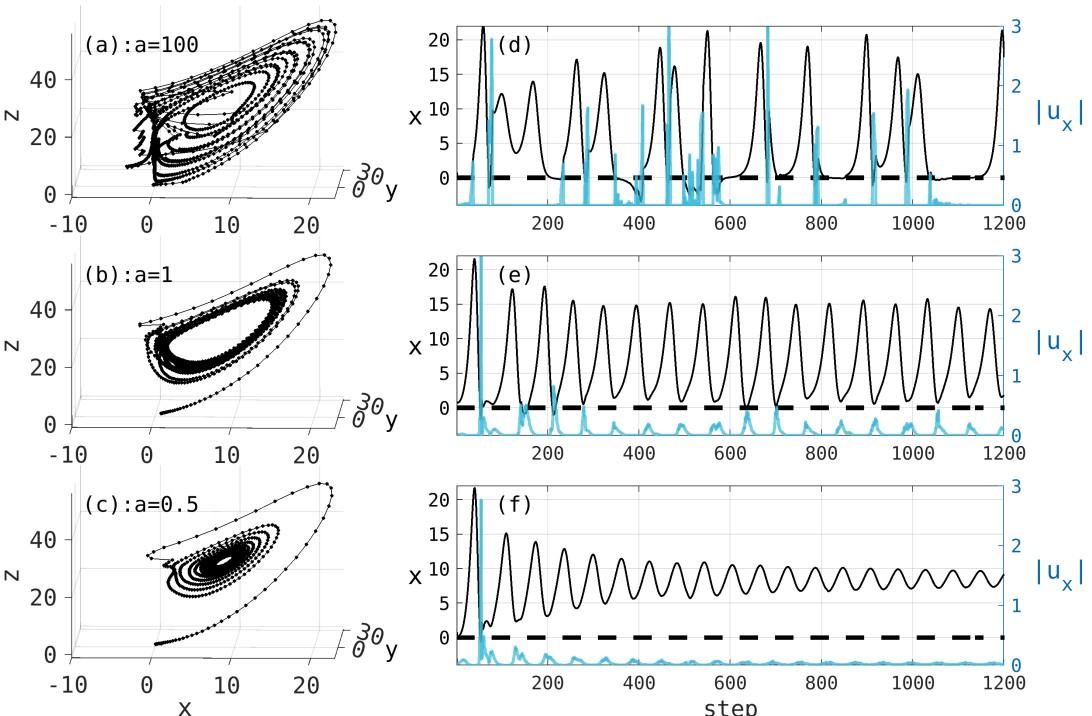

FIG. 3. Comparison of results based on different values of $a$ in the pseudo-observation operator shown in Fig. 2. The Lorenz63 model is controlled to keep the $x$ value positive, showing the behavior over the first 1200 steps. Panels (a), (b), and (c) show the attractors of the controlled NR for $a = 100$, $a = 1$, and $a = 0.5$, respectively. Panels (d), (e), and (f) show the evolution of $x$ (left axis) in the controlled NR over time, with the blue lines indicating the control inputs $|u_x|$ (right axis).

all variables, $x$, $y$, $z$. The pseudo-observation operator, $H^p$, is set to the identity operator in this approach, meaning that the pseudo-observations directly correspond to the states of the reference trajectory without additional transformations.

## 5. Experimental results

In this section, we present the experimental results evaluating the performance of the proposed EnMPC using the Lorenz63 model. We compare two approaches, the penalty term approach and the trajectory tracking approach, for the control problem of restricting the state variable $x$ to positive values. Furthermore, we examine how the choice of ensemble data assimilation methods forming



| | Approach | Estimated control inputs | Prediction horizon $T_p$ (hr) | Base DA method in EnMPC | Figure |
|---|---|---|---|---|---|
| (a) | penalty term ($\mathbf{y}^p$: $x = 0$) | $u_x$ | 48 | 4DEnVar | Fig. 3 |
| | | | | 4DEnVar, EnKS, PS | Fig. 4 |
| | | | 6, 24, 48, 120 | | Fig. 7a |
| (b) | trajectory tracking ($\mathbf{y}^p$: $x$, $y$, $z$ from ref. traj.) | $u_x, u_y, u_z$ | 48 | | Figs. 5, 6 |
| | | | 6, 24, 48, 120 | | Fig. 7b |

TABLE 1. Experimental setup

the basis of EnMPC (4DEnVar, EnKS, and PS) impacts its performance. In addition, we compare EnMPC with conventional MPC to assess its computational efficiency and control performance. Note that for the conventional MPC, we set the weighting matrix for the control input $\mathbf{C}^u$ to $0.01\mathbf{I}$, which matches the psuedo-observation error covariance matrix $\mathbf{C}^r$. We use an ensemble size of 10 for all experiments. All experiments are conducted using MATLAB on a typical laptop.

*a. Control using the penalty term approach*

In the penalty term approach, we restrict $x$ to positive values by imposing penalties on regions where $x \leq 0$. Specifically, we utilize a pseudo-observation $y^p = 0$ and a control operator $H^p(x) = \log(1 + \exp(-ax))/a$ with $a = 0.5$. In this case, we apply control only through $u_x$ using (20).

As shown in Fig. 4a, while $x$ fluctuates between positive and negative values in the NR, all four MPC methods generally restrict $x$ to the $x > 0$ region. This demonstrates that the proposed method successfully solves the MPC problem using ensemble approximations. In addition, the penalty term approach achieves control that takes into account constraint conditions by using the pseudo-observation operator.

The comparison of control inputs $u_x$ shown in Fig. 4e shows that, during the initial 400 steps, the control input for EnMPC based on PS is larger than those for the other methods (4DEnVar and EnKS). As described in Sec. 2, this is because EnKS-based and 4DEnVar-based EnMPC use ensemble-based linear transformations, which help retain the statistical structure of the original ensemble (Lorenc 2003; Poterjoy and Zhang 2015; Houtekamer and Zhang 2016; Kurosawa and Poterjoy 2023). In contrast, PS-based EnMPC determines the analysis state through resampling, where particles with higher weights are replicated while those with lower weights are removed. This can lead to the analysis state being dominated by a few specific particles, potentially causing





more abrupt changes in the control input. However, this experiment uses a nonlinear observation

operator $H^p(x) = \log(1 + \exp(-ax))/a$ as the penalty function, which posed challenges for EnKS-

based and EnVar-based EnMPC as they inherently assume Gaussianity. In contrast, PS-based

EnMPC is more appropriate for handling non-Gaussian structures and is less affected by such

assumptions (Poterjoy 2016; Poterjoy et al. 2019; Kurosawa and Poterjoy 2021).

Beyond step 400, the success rate of control approaches nearly 100% for all MPC methods,

and during this period, the magnitudes of control inputs for the three EnMPC methods show no

significant differences. This suggests that the choice of data assimilation method influences the

performance especially during the initial stages.

When comparing conventional MPC and EnMPC, it becomes clear that EnMPC achieves sig-

nificantly reduced control input magnitudes, which leads to smaller oscillations compared to

conventional MPC. This is likely because conventional MPC uses a fixed control weight matrix

$\mathbf{C}^u$ in (1), whereas EnMPC estimates it from the analysis ensemble as $\mathbf{P}^a$ in (16).

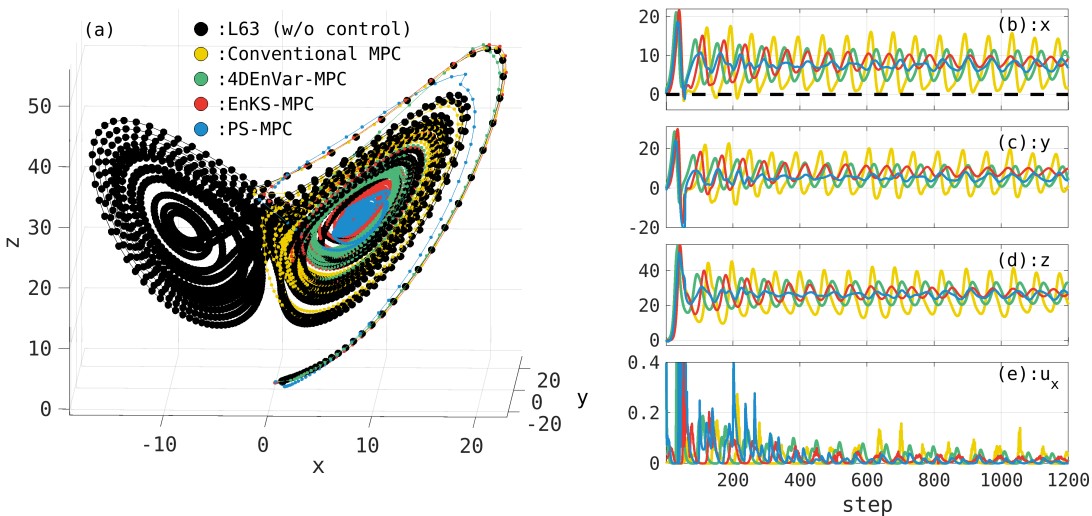

FIG. 4. Comparison of results using the conventional MPC and EnMPC with the penalty term approach. (a)

The trajectory of the uncontrolled and controlled NR, (b) time series of the values of $x$, (c) $y$, and (d) $z$ in the

controlled NR, and (e) the estimated control input $u_x$. The black dots represent the trajectory of the uncontrolled

NR, and the yellow dots show controlled NR by the conventional MPC. Green, red, and blue represent the

trajectories of the NR controlled by EnMPC based on 4DEnVar, EnKS, and PS, respectively. The dashed line in

(b) indicates the control objective, where $x > 0$.



*b. Control using the trajectory tracking approach*

The trajectory tracking approach controls the system state towards a predefined reference trajectory that satisfies $x > 0$. We employ the trajectory data from Kawasaki and Kotsuki (2024) as the reference and consider all three control variables $u_x$, $u_y$, and $u_z$ using (21).

The results demonstrate that the proposed EnMPC can accurately follow the reference trajectory. In particular, 4DEnVar-based and EnKS-based EnMPC provide smooth and stable control inputs, while PS-based EnMPC requires larger control inputs (Figs. 5e–g ). As mentioned in Sec. 5a, this is because PS-based EnMPC uses particles to represent the distribution, whereas the other two methods use ensemble-based transformations. In terms of tracking performance, PS-based EnMPC achieves significantly lower root mean squared error (RMSE) of 0.22 compared to 3.04 and 3.03 for 4DEnVar-based and EnKS-based EnMPC, respectively (Fig. 6). This suggests that the PS-based EnMPC, known for its flexibility in handling nonlinear regimes, can more accurately represent complex behaviors like the reference trajectory. In contrast, EnKS-based and EnVar-based EnMPC struggle to properly incorporate the nonlinearities of the reference trajectory, resulting in larger RMSE values.

When compared to conventional MPC, all EnMPC methods exhibit significant advantages in both tracking performance and control efficiency. Conventional MPC shows an RMSE of 5.91 (Fig. 6), which is considerably higher than any of the EnMPC methods, demonstrating its difficulty in accurately following the reference trajectory. As discussed in Sec.5a, this is likely due to the fixed control weight matrix $\mathbf{C}^u$ in conventional MPC, which limits its flexibility in adapting to the reference trajectory in the prediction horizon.

To enhance the accuracy of the control in both conventional MPC and EnMPC, or to reduce the abrupt control inputs in PS-based EnMPC, improving the prediction horizon or increasing ensemble sizes would be effective. These improvements remain an important subject for future research.

*c. Impact of prediction horizon on computational time and control performance*

This section provides a comparison of the computational time required by conventional MPC and various EnMPC methods across different prediction horizons ($T_p$). We perform the comparison for both the penalty term approach (Fig. 7a) and the trajectory tracking approach (Fig. 7b).





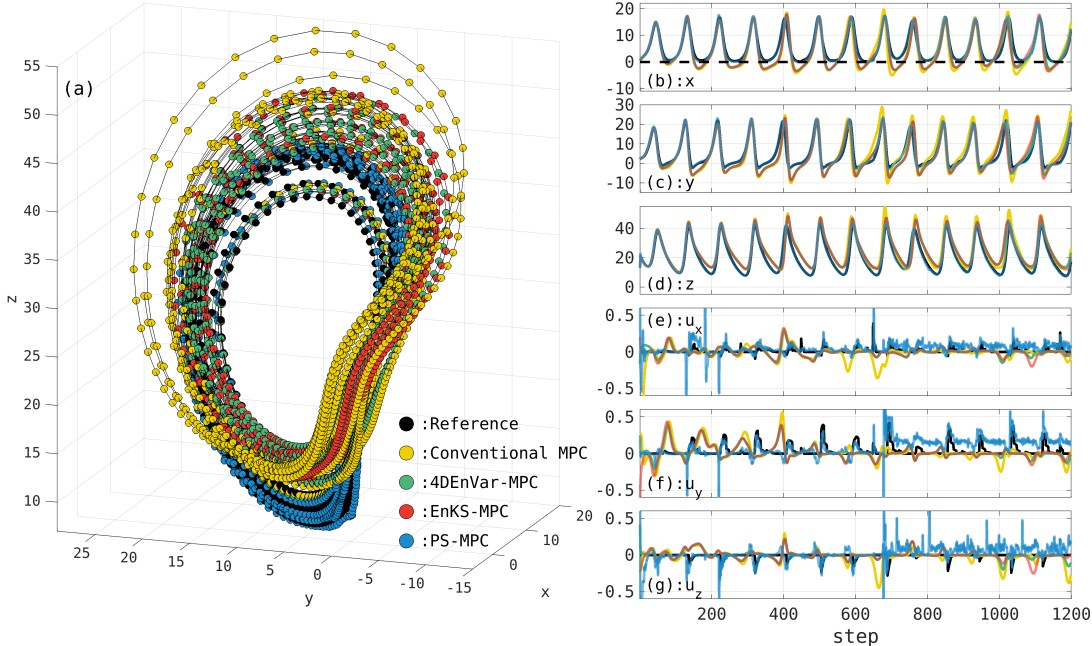

FIG. 5. As in Fig. 4, but the optimal control input values are determined to follow a reference trajectory that satisfies the constraints. The black dots represent the reference trajectory.

In the penalty term approach (Fig. 7a), EnMPC methods consistently achieve high success rates (approximately 1.0) across all prediction horizons. In contrast, conventional MPC fails to control effectively when the prediction horizon is short (6 and 24 hours). In terms of computational time, conventional MPC exhibits a sharp increase as $T_p$ extends, reflecting its computational inefficiency due to the need for full-model evaluations to calculate optimal control inputs. For example, at $T_p = 120$ hr, the computational time for conventional MPC is 620 s. On the other hand, the EnMPC methods all show a much lower computational times, with the PS-based approach yielding 121 s, the 4DEnVar-based approach 81 s, and the EnKF-based approach being the most computationally efficient at 16 s. This is because the 4DEnVar and PS methods used in the current study require iterations to determine the optimal control inputs, whereas EnKS does not. Exploring alternative data assimilation methods to further reduce computational time remains an important future research topic.





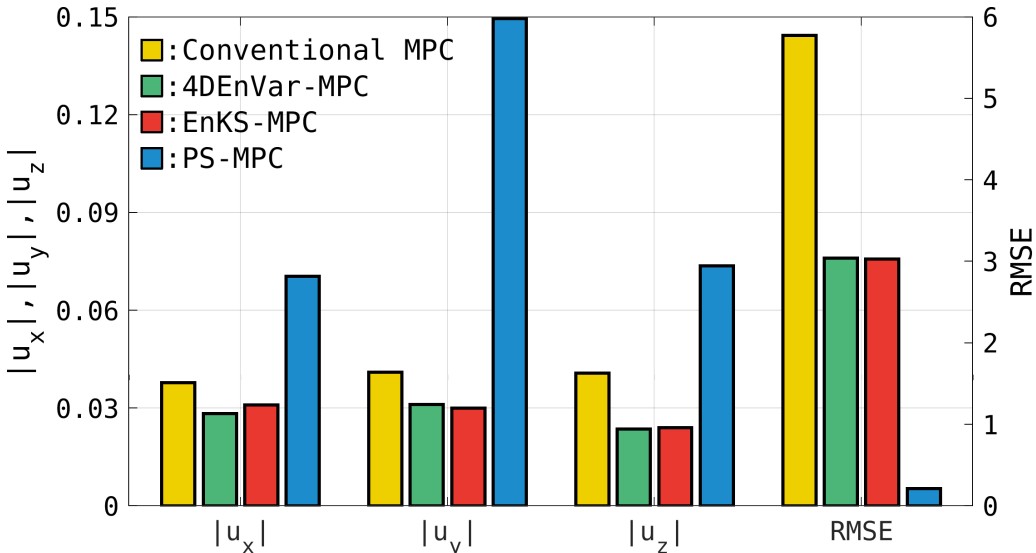

Fig. 6.    Comparison of the average control input magnitudes ($|u_x|$, $|u_y|$, and $|u_z|$; left axis) and RMSE (right axis) with respect to the reference trajectory, calculated as averages from step 400 to step 2000. Yellow, green, red, and blue bars represent conventional MPC, 4DEnVar-based EnMPC, EnKS-based EnMPC, and PS-based EnMPC, respectively. These values correspond to the results in Fig. 5.

For the trajectory tracking approach (Fig. 7b), the PS-based EnMPC achieves the lowest RMSE, maintaining high control accuracy across all prediction horizons. This is because PS does not assume Gaussianity and effectively handles the nonlinear regime, making it well-suited for accurately representing complex reference trajectories. In contrast, conventional MPC exhibits significantly higher RMSE values, indicating difficulty in tracking the reference trajectory, regardless of $T_p$. In terms of computational time, PS-based EnMPC requires slightly higher computational costs compared to other EnMPC methods, but it remains much more efficient than conventional MPC (e.g., at $T_p = 120$ hr: conventional MPC = 651 s, 4DEnVar-based = 119 s, EnKF-based = 16 s, PS-based = 158 s). This suggests that PS-based EnMPC is a strong candidate for applications where high control accuracy is prioritized. Note that the relatively higher computational cost of PS-based EnMPC in this study is due to the iterative approach used to prevent particle degeneracy (Poterjoy et al. 2019; Poterjoy 2022). Alternative PF or PS formulations may reduce computational





costs while maintaining performance (Penny and Miyoshi 2016; van Leeuwen et al. 2019; Kotsuki et al. 2022).

In summary, these results demonstrate that EnMPC outperforms conventional MPC in both computational efficiency and control performance. Particularly for longer prediction horizons, EnMPC effectively limits computational cost increases while maintaining high control accuracy.

## 6. Conclusion

The current study proposes EnMPC, a nonlinear control framework that combines MPC with ensemble data assimilation. EnMPC reduces computational cost while maintaining accurate control of nonlinear systems by using ensemble approximation. EnMPC assimilates pseudo-observations in a manner similar to actual observations in data assimilation to reflect constraints or reference trajectories of control problems. This unique approach provides an effective and flexible solution for addressing the challenges posed by complex and high-dimensional systems, such as those in meteorology and weather control.

We introduce two methods within the EnMPC framework: the penalty term approach and the trajectory tracking approach. The penalty term approach imposes penalties when the system violates constraints, ensuring the system remains within acceptable behavior. In contrast, the trajectory tracking approach guides the system to follow a pre-defined trajectory that is designed to satisfy the constraints. Both approaches demonstrate their effectiveness in controlling the chaotic dynamics of the Lorenz63 model, showing their potential to manage complex system behavior and their adaptability to diverse control objectives. The choice between these two approaches depends on the specific control problem. Selecting the appropriate method based on its characteristics and objectives is essential and remains a key area for future research.

Our experiments highlight the strengths of EnMPC compared to conventional MPC, particularly in terms of computational efficiency and flexibility. This advantage is primarily due to the fact that conventional MPC relies on the full model for optimization, whereas EnMPC uses ensemble approximations. Additionally, EnMPC determines the weights for control inputs using the analysis error covariance derived from ensemble data assimilation, while conventional MPC uses fixed control weights, limiting its adaptability to varying system dynamics.




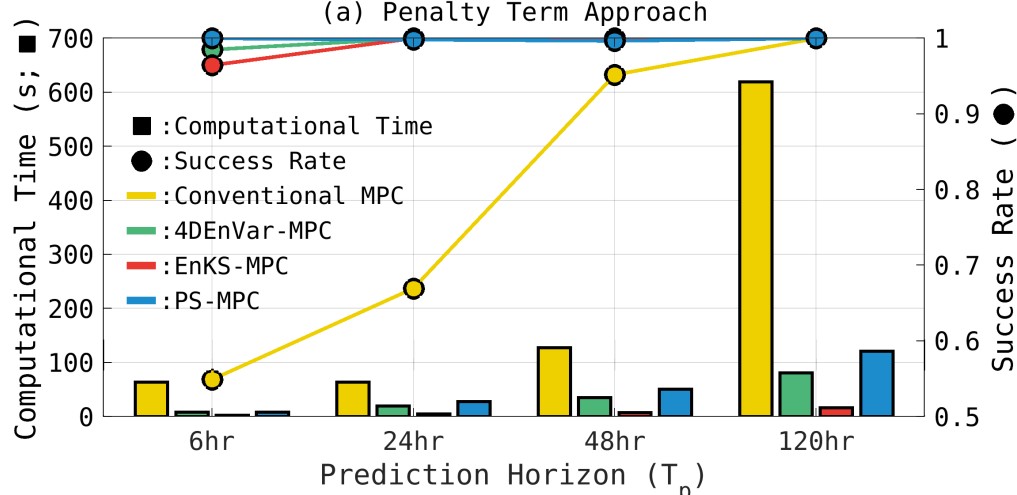

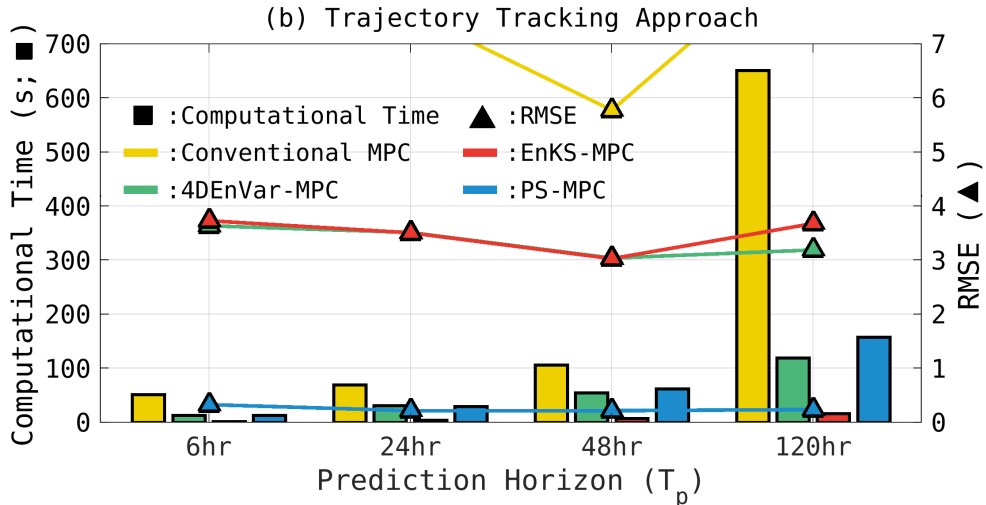

FIG. 7. Comparison of computational time and performance metrics (success rate and RMSE) as a function of the prediction horizon ($T_p$). Panel (a) shows the penalty term approach, depicting computational time (bars, left axis) and success rate (circles, right axis), where a higher success rate indicates more effective control. Panel (b) illustrates the trajectory tracking approach, highlighting computational time (bars, left axis) and RMSE (triangles, right axis), where a lower RMSE indicates more accurate tracking of the reference trajectory. Yellow, green, red, and blue bars represent conventional MPC, 4DEnVar-based EnMPC, EnKS-based EnMPC, and PS-based EnMPC, respectively. The values for $T_p = 48$ hours in panel (a) and (b) correspond to the results presented in Fig. 4 and 5, respectively.



A key aspect of our investigation involves exploring the performance of different ensemble data assimilation methods that form the foundation of the EnMPC framework, which highlights the importance of selecting the appropriate ensemble smoother method, such as 4DEnVar, EnKS, and PS. For instance, while 4DEnVar-based and EnKS-based EnMPC provide smooth and efficient control, the flexibility of PS-based EnMPC in handling nonlinear and non-Gaussian dynamics leads to greater accuracy, particularly when tracking nonlinear reference trajectories.

Despite its advantages, EnMPC is sensitive to factors such as the pseudo-observations, prediction horizon, ensemble size, and the choice of data assimilation method. For instance, achieving optimal performance with the penalty term approach requires careful tuning of pseudo-observation operators. The sensitivities highlight the need for further investigation and optimization to enhance the effectiveness and applicability of EnMPC.

In conclusion, EnMPC represents a promising framework for controlling chaotic and nonlinear systems, with potential applications extending to operational weather control. Future work will focus on addressing the remaining challenges, including improving computational efficiency, optimizing parameter selection, and mitigating sampling errors. By advancing these areas, EnMPC could become a powerful tool for operational applications, offering new possibilities for weather control and beyond.

*Acknowledgments.* This work was supported by the Japan Science and Technology Agency Moonshot R&D program Grant Numbers #JPMJMS2389-4-1, JPMJMS2389-4-2, and JSPS KAKENHI Grant Number #JP24K22969.

*Code/Data availability.* All software, documentation, and methods used to support this study are available from the corresponding author at Chiba University.

*Author contribution.* KK, AO, FK, and SK conceptualized this study. KK conducted the numerical experiments and wrote the paper. AO, FK, and SK provided comments that improved the clarity of the manuscript.

*Competing interests.* The contact author has declared that neither of the authors has any competing interests.





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
