# Peer review of "Bridging Data Assimilation and Control: Ensemble Model Predictive Control for High-Dimensional Nonlinear Systems"

_EGUsphere, 2025_

## Referee Comment (RC1)

**Report on the manuscript "Bridging Data Assimilation and Control: Ensemble Model Predictive Control for High-Dimensional Nonlinear Systems" by Kenta Kurosawa , Atsushi Okazaki , Fumitoshi Kawasaki , Shunji Kotsuk**

March 18, 2025

The paper investigates several control strategies derived from a new ensemble predictive control (EnMPC) method. Inspired by ensemble methods in data assimilation (DA), the model predictive control problem is reformulated in a way to be solved using an ensemble of model simulations instead of the standard adjoint method. In addition to the treatment of non-linearities, a strong advantage is the breaking of the sequential character of the numerical procedure, which can be crucial for real-time control or operational applications. Based on this idea, all variants of ensemble methods can be adapted to MPC, and the paper compares different classes such as ensemble Kalman filter/smoother, ensemble 4D-Var and particle filters. The comparisons are performed on the Lorentz 63 benchmark system.

I think that the study has a strong potential impact, and the comparison is novel. It is really interesting to make this comparison in order to choose one of the methods for further applications in light of their respective behaviour. However, I believe that the methodological description needs significant improvement in order to fully understand what is performed. In particular, the new EnMPC is formulated as a data assimilation problem, rather than a control problem. I suspect that this has some consequences for the results presented, which may affect their interpretation. In addition, the context and positioning of the study could be significantly improved to be fully convincing. Below are more detailed comments and questions, to be addresses before I can recommend the paper for publication.

**General comments**

I begin with comments and questions about the technical content.

- In my point of view, there is one main issue that affects the whole paper, and it is the following. In the presentation of EnMPC in section 3, the control problem is formulated as a data assimilation problem. On the one hand this simplifies the adaptation of ensemble methods to MPC, but on the other hand it leads to some inconsistencies. It is more than a simple presentation problem, because at best it requires some clarifications to understand what is really being done, and at worst it induces some inconsistencies in the results.

  First, the definition of pseudo-observations is confusing. Usually in control (often in robust optimal control) an objective output variable plays this role [see for instance Sipp and Schmid, 2016]. It would be clearer to speak of "objective output" rather than "pseudo-observations" and change the notations accordingly. This is just a matter of presentation.

  The second point is the fact that in the control problem, the variable sought is the control $\boldsymbol{u}(t)$. In equation (16) the cost functional is a function of the initial condition (as in data assimilation). In addition, the penalty term is associated with the initial condition, which is associated with the analysis error covariance matrix. Apparently, instead of penalising the control value, the initial condition anomaly is penalised. Is this just an analogy or is it really the implemented cost

functional? Is it assumed that the control is a direct forcing on the variable $x_i$ and $\mathbf{P}^a$ is used as the covariance for the control $\boldsymbol{u}$? There is a hint l.239, which states that the control inputs act only at the initial time, but it is not clear how restrictive this is. As discussed in the results, the definition of the penalty term has a strong impact and explains the better performances compared to conventional MPC.

This requires clarifications. If the control is penalised, this does not correspond to what is written, and justification of the use of $\mathbf{P}^a$ would be welcome. If there is no control penalty, this should be better justified; this would explain the RMSE performance improvement, but I do not understand the lower control values.

- How control ensemble members is sampled? Similarly as the initial conditions in the data assimilation problems? This is not clearly explained, but I think that it will become clearer after answering the first comment.

- The presentation of classical methods implemented in the results is not fully self-containing. For instance it is not told how the MPC problem is classically solved (using adjoint method I guess). The Kalman smoother variant is left to search in the cited references. Similarly, the resampling for particle filters is not explained.

- At first read, it is difficult to understand the interest of considering $T_c < T_p$ in MPC. After reading the EnMPC formalism, we understand that a 1 time step is performed in the results (See l. 239). This could be announced to avoid a pending question. Moreover, it is not clear how much the 1 time step control is required for keeping consistency in the formalism (see first comment).

- Equation (9), $\delta\boldsymbol{x}$ seems to be the ensemble anomaly, while it has been defined as the increment in incremental 4D-Var l.132.

- l.197, the presumed similarity between eq. (13) and the MPC penalty is not restricted to particle filters but all presented DA techniques. After taking into account the first comment, the similarity may be less obvious.

- l.207, It is not clear to what procedure is referred the "adaptive selection of control outputs".

- Equation (17): is the observation error $\mathbf{U}_t^y\boldsymbol{v} - \boldsymbol{d}_t^p$ or $\mathbf{U}_t^y\boldsymbol{v} - \boldsymbol{y}_t$?

- I believe that a factor 2 is missing for the gradient in equation (18).

- l.295: "choice of data assimilation". The DA are an ETKF for all tests, and it is the MPC method which is chosen.

- Interpretations l.413 about the reasons why ensemble-based linear transformations lead to smaller control is not fully clear. I believe that this interpretation should be improved in light of the first comment.

Here are questions and comments concerning the context and positioning of the study.

- In the title, the part "Bridging data assimilation and control" and the related positioning in the text is overstated in my point of view. Indeed, the data assimilation problem is related in the control community to design an estimator. The full information control problem and the full control problem are dual [Zhou et al., 1996, p. 423] and it is not surprising that they have the same structure. Afterward, they have to be coupled. In the present paper, the bridge between DA and control lies in the use of this structure similarity to solve the control problem, and in the coupling section 4.a.

- The word "High dimensional" in the title and the related positioning in the text is also exaggerated. Even if ensemble methods are designed for high-dimensional systems, the methods are applied in the paper to the Lorentz system, whose dimension is 3. Even if it has been designed for

convection cells in the atmosphere, it results from a Galerkin projection on 3 modes. In particular, the success of particle filters is not guaranteed as the dimension increases, since it is known (as mentioned in the introduction) that they are subject to degeneracy for high-dimensional systems.

- The weather control application is very restrictive compared to the potential impact of the present study. It makes feel as an ad-hoc justification to solve a control problem in the geophysical fluid dynamics community.

- An advantage of ensemble methods compared to adjoint method is that the latter requires sequential iterations between direct and adjoint model, while the former can be straightforwardly parallelised, which is very interesting for real time control and operational applications. If the authors agree, I believe that this could be mentioned.

- A better review of ensemble MPC would be welcome. A quick search lead me to the preprint [Yamaguchi and Ravela, 2023] and related conference communications. I believe that EnMPC is very new and a hot topic, with apparently few groups working on it and the review can be almost exhaustive. Moreover, similar names of EnMPC apparently refer to something else and a clear distinction may be welcome. I believe that it would be beneficial to perform a rigourous review.

- Full information control is presented in section 3, since the initial conditions are assumed to be known (up to clarifications of the first remark). The coupling between the data assimilation and the full information control is presented in section 4, but presented as "Experimental setting". This distinction may not be clear for readers not familiar with control theory. I suggest adding some remarks, or even reorganising sections to highlight the coupling, which is a very interesting feature of the paper.

- In the conclusion, the step between the Lorentz 63 system and operational systems is big. There is a range of intermediary steps and I believe that it may not be presented as a simple "extension" (l.544), but a nice mid/long-term perspective.

**Typos**

- l.75: "control objectives are unclear". To reformulate.

- l.331: space after the semi-colmn.

- l.334: "psuedo" $\mapsto$ "pseudo".

- l/356: "calculated" $\mapsto$ "applied".

**References**

D. Sipp and P. J. Schmid. Linear closed-loop control of fluid instabilities and noise-induced perturbations: A review of approaches and tools. *Applied Mechanics Reviews*, 68(2):020801, 2016.

E. Yamaguchi and S. Ravela. Multirotor ensemble model predictive control I: Simulation experiments. *arXiv preprint arXiv:2305.12625*, 2023.

K. Zhou, J. C. Doyle, K. Glover, et al. *Robust and optimal control*, volume 40. Prentice Hall New Jersey, 1996.

---

## Referee Comment (RC2)

Review of the paper:
"*Bridging Data Assimilation and Control: Ensemble Model Predictive Control for High-Dimensional Nonlinear Systems*"
by Kenta Kurosawa, Atsushi Okazaki, Fumitoshi Kawasaki and Shunji Kotsuk

March 27, 2025

**1    General comment**

The paper introduces an original framework combining control and data assimilation. The proposed method called 'ensemble model predictive control' (En-MPC) solves the MPC cost function using ensemble smoother methods such as 4DEnVar, ensemble Kalman smoother (EnKS), and particle smoother (PS). Numerical experiments are based on the Lorenz-63 model.

Despite the interesting idea of coupling control and data assimilation frameworks, the main contribution of the paper is not obvious compared to state-of-the-art methods. The new algorithm should be further detailed. Moreover, its efficiency for high-dimensional systems is not shown in the experiments.

Below are the related comments that deal with improvements, leading to a future acceptance.

**2    Major comments**

- The assertion in Abstract (Line 19) and Conclusion (Line 515) about the ability of the proposed method to handle high-dimensional systems is not verified in the numerical experiments. That is why it is going too far to mention this in the paper's title (Line 2). Note that though nonlinearity is mainly treated using the PS, the latter is not suitable for high-dimensional systems.

- Regarding e.q (17), what is the novelty compared to the works cited lines 141, 156 and 157 ?

**3   Minor comments**

- In Introduction, the difference between sequential and variational data assimilation should be more detailed (Line 73). Moreover, precise that in a variational point of view the EnKF can can be formulated as in eq. (7) (Line 160).

- What pseudo-observation means ? (Line 243)

- How the success rate is computed in Figure 7a ?

- The URL link of the paper of Fairbairn et al. (2014) is wrong, the correct one is: https://rmets.onlinelibrary.wiley.com/doi/10.1002/qj.2135

- Cite Figure 5a for ease of reference (Line 441).

- Thoughtlessness:

    - remove 'of' (Line 211)
    - index $p$ is missing on $y_t$ (Line 249)
    - replace the point after $u_x$ by a comma (Line 324)
    - write 'pseudo' (Lines 334 and 399)

---

## Author Response (AR1)

**Replies to reviewer comments**

Kenta Kurosawa, Atsushi Okazaki, Fumitoshi Kawasaki, and Shunji Kotsuki

May 29, 2025

Manuscript No.: egusphere-2025-595

Title:
↓
Ensemble-Based Model Predictive Control Using Data Assimilation Techniques

Thank you very much for your favorable evaluation of our work and insightful review comments. We provide our point-by-point responses to the comments below. Please note that our replies are indicated in blue font and that the line number of the corresponding responses may be changed because of the revision.

**Reviewer #1**

**Overview:**

The paper investigates several control strategies derived from a new ensemble predictive control (EnMPC) method. Inspired by ensemble methods in data assimilation (DA), the model predictive control problem is reformulated in a way to be solved using an ensemble of model simulations instead of the standard adjoint method. In addition to the treatment of non-linearities, a strong advantage is the breaking of the sequential character of the numerical procedure, which can be crucial for realtime control or operational applications. Based on this idea, all variants of ensemble methods can be adapted to MPC, and the paper compares different classes such as ensemble Kalman filter/smoother, ensemble 4D-Var and particle filters. The comparisons are performed on the Lorentz 63 benchmark system.
I think that the study has a strong potential impact, and the comparison is novel. It is really interesting to make this comparison in order to choose one of the methods for further applications in light of their respective behaviour. However, I believe that the methodological description needs significant improvement in order to fully understand what is performed. In particular, the new EnMPC is formulated as a data assimilation problem, rather than a control problem. I suspect that this has some consequences for the results presented, which may affect their interpretation. In addition, the context and positioning of the study could

be significantly improved to be fully convincing. Below are more detailed comments and questions, to be addresses before I can recommend the paper for publication.

**General Comments:**

1. In my point of view, there is one main issue that affects the whole paper and it is the following. In the presentation of EnMPC in section 3, the control problem is formulated as a data assimilation problem. On the one hand this simplifies the adaptation of ensemble methods to MPC, but on the other hand it leads to some inconsistencies. It is more than a simple presentation problem, because at best it requires some clarifications to understand what is really being done, and at worst it induces some inconsistencies in the results.

   First, the definition of pseudo-observations is confusing. Usually in control (often in robust optimal control) an objective output variable plays this role [see for instance Sipp and Schmid, 2016]. It would be clearer to speak of "objective output" rather than "pseudo-observations" and change the notations accordingly. This is just a matter of presentation. The second point is the fact that in the control problem, the variable sought is the control u(t). In equation (16) the cost functional is a function of the initial condition (as in data assimilation).

   In addition, the penalty term is associated with the initial condition, which is associated with the analysis error covariance matrix. Apparently, instead of penalising the control value, the initial condition anomaly is penalised. Is this just an analogy or is it really the implemented cost functional? Is it assumed that the control is a direct forcing on the variable xi and Pa is used as the covariance for the control u? There is a hint l.239, which states that the control inputs act only at the initial time, but it is not clear how restrictive this is. As discussed in the results, the definition of the penalty term has a strong impact and explains the better performances compared to conventional MPC.

   This requires clarifications. If the control is penalised, this does not correspond to what is written, and justification of the use of Pa would be welcome. If there is no control penalty, this should be better justified; this would explain the RMSE performance improvement, but I do not understand the lower control values.

   Reply:
   Thank you for your insightful comments. We have replaced "pseudo-observations" with "objective outputs" throughout the manuscript.

   We revised the description associated with Equation (16) as follows (Line 117-118 and 263–267):

   "In this study, the control horizon $T_c$ is shorter than the prediction horizon $T_p$, where control is applied only at the first time step of each cycle."

"In (16), the variable $\mathbf{x}_0$ is optimized as the control input to guide the system's trajectory $\mathbf{x}_t$ toward a set of desirable future states $\mathbf{y}_t^r$. Deviations of $\mathbf{x}_0$ from the initial analysis $\overline{\mathbf{x}_0^a}$, obtained via ensemble data assimilation are penalized to ensure that the control input remains realistic. Once the optimal control input $\mathbf{x}_0^*$ is found, the resulting trajectory $\mathbf{x}^c = \arg\min J(\mathbf{x}_0)$ is regarded as the controlled state."

This new explanation also clarifies the interpretation of the penalty term. Rather than applying a direct penalty on a traditional control variable $u(t)$, we assume that control acts through the initial state $\mathbf{x}_0$, and the penalty is applied to its deviation from the analysis. This setting allows us to reuse ensemble-based data assimilation frameworks while maintaining a control perspective.

2. How control ensemble members is sampled? Similarly as the initial conditions in the data as- similation problems? This is not clearly explained, but I think that it will become clearer after answering the first comment.

   Reply:
   To clarify this, we have added the following sentence in Lines 307–309 of the revised manuscript:

   "The ensemble used in the control problem is the analysis ensemble obtained through data assimilation. This ensemble reflects the flow-dependent uncertainty at the initial time and is directly employed for estimating the optimal control inputs. No additional sampling is performed specifically for control."

3. The presentation of classical methods implemented in the results is not fully self-containing. For instance it is not told how the MPC problem is classically solved (using adjoint method I guess). The Kalman smoother variant is left to search in the cited references. Similarly, the resampling for particle filters is not explained.

   Reply:
   To address this, we have made the following additions in the revised manuscript:
   Lines 119–124: We added a general description of conventional MPC, including the use of adjoint methods, as follows:

   "In conventional MPC, optimal control inputs are typically obtained by minimizing a cost function through gradient-based optimization. For nonlinear systems, this often involves solving the adjoint equations to efficiently compute gradients of the cost function with respect to control variables. Although this approach is accurate, it requires derivation and implementation of the adjoint model, which can be costly and challenging, especially for high-dimensional systems such as numerical weather prediction models."

   Lines 195–200: To clarify our implementation of the Kalman smoother variant, we added:

"In this study, we employ 4D-ETKF as our implementation of EnKS. 4D-ETKF estimates the initial state by assimilating observations distributed over a finite time window, using an ensemble-based transformation that minimizes the analysis error covariance. Unlike the original EnKS that relies on sequential updates, 4D-ETKF applies a single batch update by linearly combining ensemble perturbations, ensuring consistency and computational efficiency without the need for adjoint models."

Lines 220–224: We included a description of the resampling procedure used in the particle smoother:

"To better approximate the posterior distribution and mitigate degeneracy, where some particles have negligible weights, a resampling step is performed. During resampling, particles with higher weights are replicated, while those with lower weights are discarded, ensuring the ensemble remains focused on the most likely regions of the state space."

4. At first read, it is difficult to understand the interest of considering Tc ¡ Tp in MPC. After reading the EnMPC formalism, we understand that a 1 time step is performed in the results (See l. 239). This could be announced to avoid a pending question. Moreover, it is not clear how much the 1 time step control is required for keeping consistency in the formalism (see first comment).

Reply:
Thank you for your comment. Optimizing only the initial state $x_o$ allows us to reuse data assimilation techniques without introducing time-dependent control variables. This simplifies the implementation and reduces computational cost. While not strictly required, this setting is practical and aligns with the receding-horizon nature of MPC.

We provided additional sentences to address the reviewer's comment: (lines 117-118)

"In this study, the control horizon $T_c$ is shorter than the prediction horizon $T_p$, where control is applied only at the first time step of each cycle."

5. Equation (9), $\delta x$ seems to be the ensemble anomaly, while it has been defined as the increment in incremental 4D-Var l.132.

Reply:
To clarify this, we have added the following sentence in Line 180 of the revised manuscript:

"$\delta \mathbf{x}^{(k)}$ is the $k$-th ensemble perturbations for the model state."

6. l.197, the presumed similarity between eq. (13) and the MPC penalty is not restricted to particle filters but all presented DA techniques. After taking into account the firstc omment, the similarity may be less obvious.

Reply:
Thank you for the valuable comment. We agree that the similarity between Equation (13) and the penalty terms in MPC is not limited to particle filters, but is in fact shared across various data assimilation techniques.

To acknowledge this, we have added the following clarification in Lines 211–213:

"Although the likelihood function used in PF resembles the observation term in the cost functions of other data assimilation methods, it plays a more central and explicit role in PF."

7. l.207, It is not clear to what procedure is referred the "adaptive selection of control outputs".

Reply:
To avoid confusion, we have removed the phrase "adaptive selection of control outputs" from the manuscript.

8. Equation (17): is the observation error $\mathbf{U}_t^y \mathbf{v} - \mathbf{d}_t^r$ or $\mathbf{U}_t^y \mathbf{v} - \mathbf{y}_t$?

Reply:
The term $\mathbf{U}_t^y \mathbf{v} - \mathbf{d}_t^r$ is not a difference between two innovations. Rather, $\mathbf{d}_t^r$ itself is an innovation-like quantity defined as

$$\mathbf{d}_t^r = \mathbf{y}_t^r - H^r[M_t(\mathbf{x}_0^a)],$$

which represents the deviation of the reference output from the forecast without control. The term $\mathbf{U}_t^y \mathbf{v}$ represents the modification to the forecast induced by the control input $\mathbf{v}$ in the ensemble space. Therefore, $\mathbf{U}_t^y \mathbf{v} - \mathbf{d}_t^r$ expresses the mismatch between the controlled forecast and the reference target, and should not be interpreted as a difference between two innovation vectors.

9. I believe that a factor 2 is missing for the gradient in equation (18).

Reply:
We fixed the typo.

10. l.295: "choice of data assimilation". The DA are an ETKF for all tests, and it is the MPC method which is chosen.

Reply:
We have revised the sentence.

11. Interpretations l.413 about the reasons why ensemble-based linear trans formations lead to smaller control is not fully clear. I believe that this interpretation should be improved in light of the first comment.

Reply:
We provided additional sentences to address the reviewer's comment: (lines 444-448)

"Specifically, when the cost function includes a penalty term weighted by the inverse of the ensemble covariance, the solution is guided toward regions of high ensemble density. This acts as a form of regularization, effectively constraining the solution to subspaces spanned by the dominant ensemble modes and scaling it according to ensemble uncertainty. Compared to approaches that do not explicitly incorporate such statistical information, this often results in smaller and more dynamically consistent control inputs."

12. In the title, the part "Bridging data assimilation and control" and the related positioning in the text is overstated in my point of view. Indeed, the data assimilation problem is related in the control community to design an estimator. The full information control problem and the full control problem are dual [Zhou et al., 1996, p. 423] and it is not surprising that they have the same structure. Afterward, they have to be coupled. In the present paper, the bridge between DA and control lies in the use of this structure similarity to solve the control problem, and in the coupling section 4.a.

Reply:
Thank you for your insightful comment.

We have revised the manuscript text for clarity and changed the title to:
"Ensemble-Based Model Predictive Control Using Data Assimilation Techniques"

We revised expressions throughout the manuscript to avoid overstatements, replacing "bridging" with more accurate phrases such as "leveraging structural similarities" and "utilizing data assimilation techniques for control".

We explicitly cited Zhou et al. (1996) to acknowledge the theoretical foundations of this duality in Section 3 (Line 240-243).
"The structural similarity between estimation and control has been well established in control theory, where the full information control problem and the state estimation problem are known to be duals Zhou et al. (1996)."

The heading of Section 4.a was revised to "Coupling of Data Assimilation and Control" to make this connection more transparent to the reader.

13. The word "High dimensional" in the title and the related positioning in the text is also exag- gerated. Even if ensemble methods are designed for high-dimensional systems, the methods are applied in the paper to the Lorentz system, whose dimension is 3. Even if it has been designed for convection cells in the atmosphere, it results from a Galerkin projection on 3 modes. In particular, the success of particle filters is not guaranteed as the dimension increases, since it is known (as mentioned in the introduction) that they are subject to degeneracy for high-dimensional systems.

Reply:
Thank you for this important comment. To address this, we have revised the title.
In addition, we carefully revised the wording throughout the manuscript to avoid exaggerating the method's suitability for high-dimensional systems. We now present such claims more cautiously, especially in the abstract, introduction, and conclusion.

While particle filters are known to suffer from degeneracy in high-dimensional settings, we note that localization strategies have been developed to mitigate this issue (e.g., Poterjoy 2016; Penny and Miyoshi 2016). We have added a sentence to clarify that such techniques may enable the practical application of our framework to more complex systems (lines 585-588).
" In particular, ensemble methods including particle filters can be adapted to higher-dimensional settings by introducing localization techniques, as demonstrated in prior data assimilation studies. While particle filters face challenges such as degeneracy in high-dimensional spaces, recent advances in localized and hybrid particle filter approaches offer promising directions for overcoming these limitations."

14. The weather control application is very restrictive compared to the potential impact of the present study. It makes feel as an ad-hoc justification to solve a control problem in the geophysical fluid dynamics community.

    Reply:
    To clarify this point, we have added the following sentences in Lines 585–589 of the revised manuscript:

    "These include not only operational weather models but also other nonlinear dynamical systems such as ocean circulation models, ecosystem dynamics, and economic or neural systems. Addressing key challenges—such as improving computational efficiency, optimizing parameter selection, and mitigating sampling errors—will be essential for these extensions."

15. An advantage of ensemble methods compared to adjoint method is that the latter requires se- quential iterations between direct and adjoint model, while the former can be straightforwardly parallelised, which is very interesting for real time control and operational applications. If the authors agree, I believe that this could be mentioned.

    Reply:
    To clarify this point, we have added the following sentences in Lines 119–124 of the revised manuscript:
    "In conventional MPC, optimal control inputs are typically obtained by minimizing a cost function through gradient-based optimization. For nonlinear systems, this often involves solving the adjoint equations to efficiently compute gradients of the cost function with respect to control variables. Although this approach is accurate, it requires

derivation and implementation of the adjoint model, which can be costly and challenging, especially for high-dimensional systems such as numerical weather prediction models."

We have added the following sentences in Lines 277–281 of the revised manuscript:
"One key advantage of ensemble-based methods over adjoint-based approaches is their suitability for parallel computation. Adjoint methods require sequential iterations between forward and backward (adjoint) models, which can be computationally demanding and less scalable. In contrast, ensemble methods allow for straightforward parallelization across ensemble members, making them highly attractive for real-time control and operational applications."

16. A better review of ensemble MPC would be welcome. A quick search lead me to the preprint [Yamaguchi and Ravela, 2023] and related conference communications. I believe that EnMPC is very new and a hot topic, with apparently few groups working on it and the review can be almost exhaustive. Moreover, similar names of EnMPC apparently refer to something else and a clear distinction may be welcome. I believe that it would be beneficial to perform a rigourous review.

Reply:
We have added the following sentences in Lines 87–93 of the revised manuscript:
"Yamaguchi and Ravela, (2023) proposed an ensemble MPC framework using fully nonlinear forward simulations and Gaussian processes for backward gain computation. While their approach is innovative and effective for control in low-dimensional robotic systems, our proposed EnMPC framework differs in several key aspects. Specifically, we integrate ensemble-based data assimilation techniques into the control framework, allowing the assimilation of actual observations and the estimation of both the initial state and control variables. Moreover, our focus is on high-dimensional geophysical systems, where observation-based state estimation is indispensable."

17. Full information control is presented in section 3, since the initial conditions are assumed to be known (up to clarifications of the first remark). The coupling between the data assimilation and the full information control is presented in section 4, but presented as "Experimental setting". This distinction may not be clear for readers not familiar with control theory. I suggest adding some remarks, or even reorganising sections to highlight the coupling, which is a very interesting feature of the paper.

Reply:
We have added the following sentences in Lines 295–298 of the revised manuscript:
"While Sec. 3 introduces the full information control assuming a known initial state, this section presents a more realistic setting where the initial condition is unknown and must be estimated using data assimilation."

18. In the conclusion, the step between the Lorentz 63 system and operational systems is

big. There is a range of intermediary steps and I believe that it may not be presented as a simple "extension" (l.544), but a nice mid/long-term perspective.

Reply:
We have revised the manuscript to present such claims more cautiously, particularly in the abstract, introduction, and conclusion. Please refer to our response to Comment #13 for details on these revisions.

19. (a) l.75: "control objectives are unclear". To reformulate.

    (b) l.331: space after the semi-colmn.

    (c) l.334: "psuedo" →"pseudo".

    (d) l/356: "calculated" →"applied".

Reply:
We have corrected all of the noted typos and formatting problems.

**Reviewer #2**

**Overview:**

The paper introduces an original framework combining control and data assimilation. The proposed method called 'ensemble model predictive control' (EnMPC) solves the MPC cost function using ensemble smoother methods such as 4DEnVar, ensemble Kalman smoother (EnKS), and particle smoother (PS). Numerical experiments are based on the Lorenz-63 model. Despite the interesting idea of coupling control and data assimilation frameworks, the main contribution of the paper is not obvious compared to state-of-the-art methods. The new algorithm should be further detailed. Moreover, its efficiency for high-dimensional systems is not shown in the experiments. Below are the related comments that deal with improvements, leading to a future acceptance

**Comments:**

1. The assertion in Abstract (Line 19) and Conclusion (Line 515) about the ability of the proposed method to handle high-dimensional systems is not verified in the numerical experiments. That is why it is going too far to mention this in the paper's title (Line 2). Note that though nonlinearity is mainly treated using the PS, the latter is not suitable for high-dimensional systems.

Reply:
Thank you for this important comment. To address this, we have revised the title. In addition, we carefully revised the wording throughout the manuscript to avoid exaggerating the method's suitability for high-dimensional systems. We now present such claims more cautiously, especially in the abstract, introduction, and conclusion.

2. Regarding e.q (17), what is the novelty compared to the works cited lines 141, 156 and 157 ?

Reply:
To clarify this, we have added the following sentence in Lines 266–270 of the revised manuscript:
"In (16), the variable $\mathbf{x}_0$ is optimized as the control input to guide the system's trajectory $\mathbf{x}_t$ toward a set of desirable future states $\mathbf{y}_t^{\mathbf{r}}$. Deviations of $\mathbf{x}_0$ from the initial analysis $\overline{\mathbf{x}_0^a}$, obtained via ensemble data assimilation are penalized to ensure that the control input remains realistic. Once the optimal control input $\mathbf{x}_0^*$ is found, the resulting trajectory $\mathbf{x}^c = \arg\min J(\mathbf{x}_0)$ is regarded as the controlled state."

We have added the following sentence in Lines 277–281 of the revised manuscript:
"One key advantage of ensemble-based methods over adjoint-based approaches is their suitability for parallel computation. Adjoint methods require sequential iterations between forward and backward (adjoint) models, which can be computationally demanding and less scalable. In contrast, ensemble methods allow for straightforward parallelization across ensemble members, making them highly attractive for real-time control and operational applications."

**Minor Comments:**

1. In Introduction, the difference between sequential and variational data assimilation should be more detailed (Line 73). Moreover, precise that in a variational point of view the EnKF can can be formulated as in eq. (7) (Line 160).

Reply:
Thank you for your comment. To address it, we made the following revisions:

We expanded the description of the differences between sequential and variational data assimilation methods in the introduction (Line 193-196).
"Sequential methods, such as the EnKF, update the state estimate as new observations become available, typically using a forecast–analysis cycle. In contrast, variational methods formulate the state estimation as an optimization problem over a time window, where the model trajectory is adjusted to minimize a cost function based on observations and prior estimates."

We also clarified that the EnKF can be interpreted in a variational framework, and specifically noted its connection to Equation (7) (Line 175-178).
"From a variational perspective, ensemble methods like the EnKF can be interpreted as approximating the solution to a variational cost function such as Equation (2), using ensemble statistics to represent background error covariances."

2. What pseudo-observation means ?

Reply:
To avoid confusion, all instances of "pseudo-observation" have been replaced with "objective output" throughout the manuscript.

3. How the success rate is computed in Figure 7a ?

Reply:
We have added the following sentence in Lines 509–511 of the revised manuscript: "The success rate in Fig. 7a is computed as the proportion of time steps—excluding the initial 200 spin-up steps—during which the value of $x$ remains positive."

4. The URL link of the paper of Fairbairn et al. (2014) is wrong.

Reply:
We fixed the typo.

5. Cite Figure 5a for ease of reference (Line 441).

Reply:
We have now cited Figure 5a in Line 480.

6. (a) remove 'of' (Line 211)

   (b) index p is missing on yt (Line 249)

   (c) replace the point after ux by a comma (Line 324)

   (d) write 'pseudo' (Lines 334 and 399)

Reply:
We have corrected all of the noted typos and formatting problems.